# Complex subsets but redundant clonality after B cells egress from spontaneous germinal centers

**Carlos Castrillon[1]\*[‡], Lea Simoni[1†], Theo van den Broek[1†], Cees van der Poel[1†], Elliot H Akama-Garren[1,2], Minghe Ma[1], Michael C Carroll[1,3]\***

[1]Program in Cellular and Molecular Medicine, Boston Children's Hospital, Boston, United States; [2]Harvard-MIT Health Sciences and Technology, Harvard Medical School, Boston, United States; [3]Department of Pediatrics, Harvard Medical School, Boston, United States

**Abstract** Affinity matured self-reactive antibodies are found in autoimmune diseases like systemic lupus erythematous. Here, we used fate-mapping reporter mice and single-cell transcriptomics coupled to antibody repertoire analysis to characterize the post-germinal center (GC) B cell compartment in a new mouse model of autoimmunity. Antibody-secreting cells (ASCs) and memory B cells (MemBs) from spontaneous GCs grouped into multiple subclusters. ASCs matured into two terminal clusters, with distinct secretion, antibody repertoire and metabolic profiles. MemBs contained FCRL5+ and CD23+ subsets, with different in vivo localization in the spleen. GC-derived FCRL5+ MemBs share transcriptomic and repertoire properties with atypical B cells found in aging and infection and localize to the marginal zone, suggesting a similar contribution to recall responses. While transcriptomically diverse, ASC and MemB subsets maintained an underlying clonal redundancy. Therefore, self-reactive clones could escape subset-targeting therapy by perpetuation of self-reactivity in distinct subsets.

**\*For correspondence:**
castrilloncarlos@gmail.com (CC);
michael.carroll@childrens.harvard.edu (MCC)

[†]These authors contributed equally to this work

**Present address:** [‡]Department of Medicine, Division of Rheumatology, Emory University, Atlanta, United States

**Competing interest:** The authors declare that no competing interests exist.

## Editor's evaluation

Understanding the heterogeneity of the B cell response induced in autoimmune individuals is important for the development of therapies designed to target the cells underlying disease progression. Here, the authors use a new model of autoimmunity to assess the heterogeneity of the B cell response using scRNA-seq and scBCR-seq and found that B cell responses are similar to those by exogenous protein immunization.

## Introduction

In autoimmune diseases like systemic lupus erythematosus (SLE), pathogenesis is caused by self-reactive antibody-secreting cells (ASCs) and sustained by memory B cells (MemBs). Numerous subsets of these B cells have been described in models of health and disease, but it is unknown whether and how these subsets change and relate to each other in autoimmune disease. Among MemBs, the major known subsets are conventional germinal center (GC)- and non-GC-derived MemBs (*Taylor et al., 2012*; *Viant et al., 2021*). In addition, atypical MemBs (AtMemBs) are present in models of acute and chronic infection in mice (*Kim et al., 2019*; *Song et al., 2022*), as well as in parasitic infection in humans (*Portugal et al., 2017*) and are similar to the age-associated B cells first described in ageing and autoimmune mouse models (*Phalke and Marrack, 2018*). DN2 (IgD−CD27−) cells resemble AtMemB cells and are an expanded subset of B cells in human SLE. They likely have a naive

B cell origin, and data suggest they can be readily activated to mature into ASCs in the extrafollicular space (*Jenks et al., 2018*). Indeed, these subsets of B cells maintain similarities to marginal zone (MZ) B cells, including the capacity to differentiate into ASCs. However, it is unclear how diverse these distinct subsets are and how clonally related they might be.

ASCs can also segregate into different subsets with specific properties related to lifespan (short- and long-lived) (*Nutt et al., 2015*; *Radbruch et al., 2006*) and origin (GC- and EF-derived) (*Elsner and Shlomchik, 2020*), though data on the differences in the metabolic profile and clonal overlap between these populations are still lacking. Importantly, experiments with LPS-driven derivation of B cells into ASCs have demonstrated the existence of multiple clusters of ASCs as they mature into alternate terminal states (*Scharer et al., 2020*). Overall, the complexity of the ASC compartment remains underexplored.

In SLE, isotype-class switching and somatic hypermutations of autoreactive antibodies indicate antibody maturation in GCs (*Cappione et al., 2005*). Therefore, to investigate the post-GC subpopulations of B cells in SLE, we utilized a chimeric mouse model where self-reactive clones, which are derived from B cells with a wild-type (WT) B cell receptor (BCR) repertoire, compete in spontaneous GCs (*Degn et al., 2017*). We used fate mapping and single-cell RNA sequencing to gain insight into the post-GC populations of B cells in both autoimmune chimeras and immunized chimeras. We found a diversity of ASCs at different maturation stages with distinct terminal states and multiple MemB cell clusters, including GC-derived FCRL5+ and CD23+ MemB cells. These ASCs and MemB subsets showed specific transcriptomic profiles, antibody repertoire profiles, and localization, suggesting functional diversity yet maintaining redundant clonality. Together, our findings outline the underlying complexity of the MemB and ASC compartments and indicates that therapeutic efforts to target autoreactive B cells must consider the likely reservoirs of self-reactivity in other subsets.

## Results
### Fate-mapped B cells are distributed into four main clusters
To track WT B cells as they develop in an autoimmune environment, we used a mixed bone marrow (BM) chimera model (*Akama-Garren et al., 2021*; *Degn et al., 2017*; *van der Poel et al., 2019*). Irradiated host mice received a combination of BM donor cells from WT and 564 Igi mice (*Berland et al., 2006*), which bear transgenic BCRs with anti-RNP specificities. In this model, WT donor B cells expand, are selected into spontaneous GCs, mature into ASCs, and contribute to the circulation of self-reactive antibodies. To track activated B cells and their derived populations, Aicda-CreERT2-EYFP fate-mapping reporter mice (*Dogan et al., 2009*) were used as donors for the WT B cell repertoire (*Figure 1A*). Fate-mapped WT B cells were purified, sorted, and processed for droplet-based single-cell RNA-seq. Moreover, we compared the response to that of foreign antigen WT chimeras immunized with a primary and secondary dose of haptenated protein.

A total of 12,839 cells (9535 from 5 autoimmune chimeras, 3304 from 3 immunized chimeras) were retained post quality control for further analysis. Expression of the EYFP reporter transcripts was confirmed. Using unsupervised clustering, the cells grouped into four clusters (*Figure 1B*), representing the major known B cell compartments: GC B cells (e.g. *S1pr2*), which are either dark zone (DZ, e.g. *Mki67*) or light zone (LZ, e.g. *Cd83*), ASCs (e.g. *Xbp1*), and MemBs (e.g. *Sell*) (*Figure 1C*, *Figure 1—figure supplement 1A*). All compartments, LZ, DZ, MemB, and ASC clusters were clearly represented in both conditions (*Figure 1D*).

Using paired single-cell BCR sequencing and repertoire analysis, we observed clones with clear expansion in both autoimmune and immunized chimeras (*Figure 1—figure supplement 1B*). Indeed, in both conditions, we were able to find clones that had members belonging to all four clusters, exemplified by the phylogenetics of the most expanded clone in one autoimmune chimeric mouse (*Figure 1E*). The GC mutational level was in a similar range between conditions, although autoimmune chimeras had more DZ and LZ replacement mutations. MemBs in autoimmune and immunized chimeras had a similar number of nucleotide replacement mutations. However, ASCs accumulated more mutations in immunized chimeras (*Figure 1F*). Nevertheless, ASCs in both conditions reached similar maximum levels of mutations (20 nt in autoimmune and 19 nt in immunized chimeras).

Cells from autoimmune chimeras showed more isotype diversification in all compartments (*Figure 1G*). Importantly, both preferential and shared clonal usage of heavy chain genes were observed

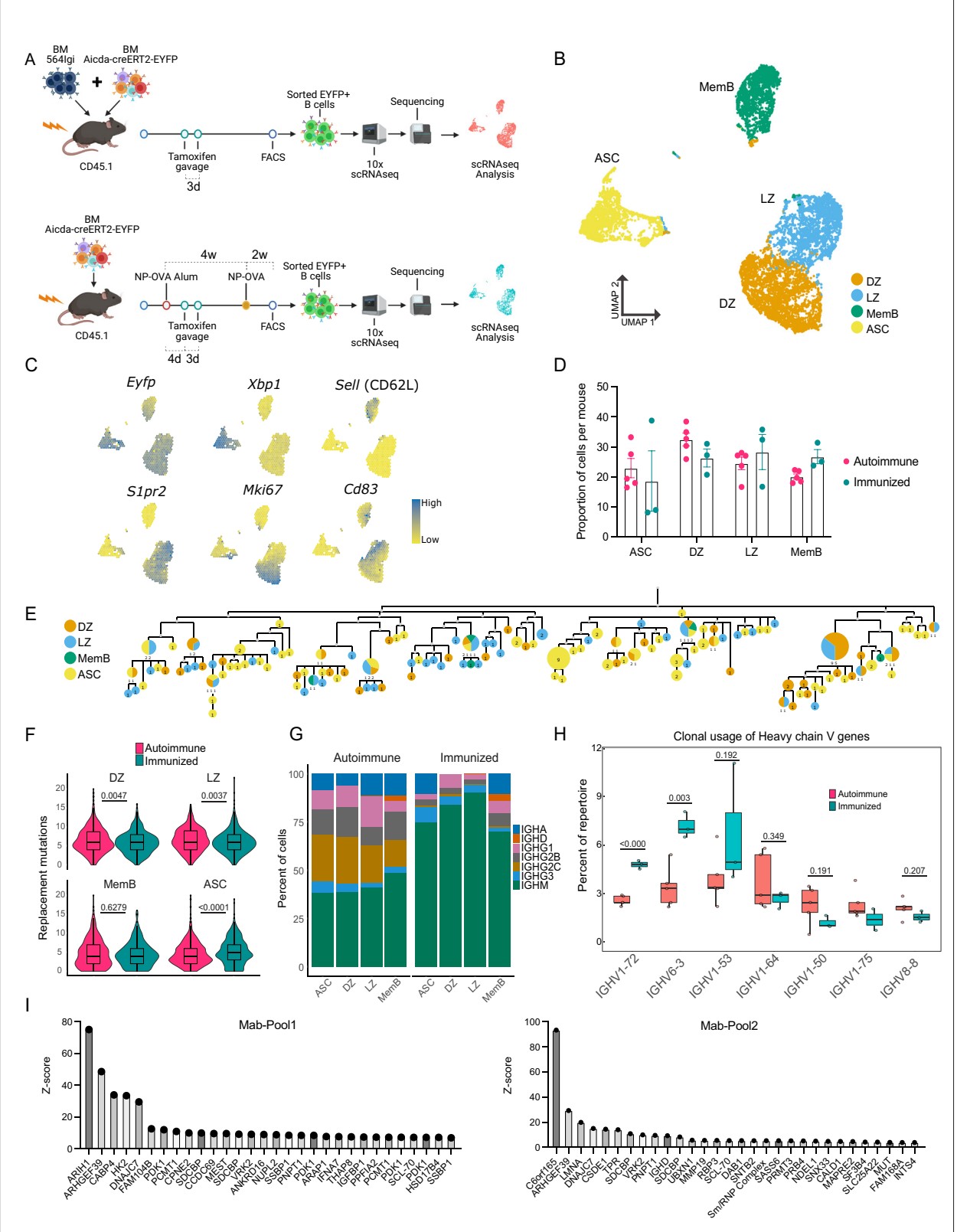

**Figure 1.** B cells with a WT repertoire are found in all compartments of autoimmune chimeras. (**A**) Experimental setup to produce the autoimmune and immunized chimeras and selection for single-cell sequencing. (**B–H**) Single-cell RNA-seq data from autoimmune ($n$ = 5) and immunized ($n$ = 3) chimeras. (**B**) Two-dimensional UMAP representation of single-cell transcriptomic data from all cells from autoimmune and immunized chimeras. (**C**) Gene expression of conventional markers was used to assign an identity to the clusters. Log normalized expression. (**D**) Proportion of cells belonging to the

*Figure 1 continued on next page*

*Figure 1 continued*

antibody-secreting cell (ASC), DZ, LZ, and memory B cell (MemB) clusters among all cells sequenced from each chimera. Each dot represents a mouse. (**E**) Phylogenetic tree of the most expanded clone in one of the autoimmune chimeras. Clonal members colored by assigned cluster. (**F**) Accumulated replacement mutations in each cluster by condition. Violin plot with embedded Tukey boxplot. (**G**) Isotype usage per cluster for each condition. (**H**) Clonal usage of selected V genes by condition. Each dot represents a mouse in a Tukey boxplot. IGHV 1–75 was only found in two immunized chimeras. (**I**) Z-score ranking of fluorescent detection for monoclonal pools Mab1 (left) and Mab2 (right) after HuProt microarray binding assay. The monoclonals originate from two autoimmune chimeras. Statistical values correspond to two-tailed Mann–Whitney test (**F**) and unpaired *t*-tests (**H**).

The online version of this article includes the following source data and figure supplement(s) for figure 1:

**Figure supplement 1.** Transcriptomic and repertoire profile of main clusters.

**Source data 1.** Data for *Figure 1D*.

**Source data 2.** Data for *Figure 1F*.

**Source data 3.** Data for *Figure 1H*.

among the most used V genes in each condition. Seven V genes were shared among the top 10 V genes in autoimmune (*Figure 1—figure supplement 1C*) and immunized chimeras (*Figure 1—figure supplement 1D*): IGHVs 1–26, 1–53, 1–64, 1–72, 3–6, 6–3, and 9–3. V genes traditionally associated with nitro-phenyl (NP) specificity, like IGHVs 1–72 (a.k.a. V168) 6–3 and 1–53 (*Xue et al., 2019*), were more expanded upon immunization. On the other hand, autoimmune chimeras preferred IGHVs 1–50, 1–75, and 8–8, whereas IGHV 1–64 was used at a similar proportion in both conditions (*Figure 1H*). Notably, commonly used V genes in both conditions had clonal usage frequency in a similar range to that observed in B cells from naïve mice (*Figure 1—figure supplement 1E*; *Rettig et al., 2018*). IGHV genes 1–72, 1–55, 1–64, 1–53, and 1–50 sequences are among the most closely related to that of the 564 Igl autoantibody (*Figure 1—figure supplement 1F*).

To investigate the potential targets, we produced six monoclonal antibodies using the VDJ sequences from two autoimmune chimeric mice. The VDJ sequences originated from members of four different clones, including the most expanded clone from one mouse. We tested the reactivity of these antibodies using a native protein HuProt array containing more than 20,000 recombinant proteins. We combined the monoclonal antibodies into two pools (Mab1 and Mab2), each containing two distinct clones. We identified multiple potential binders, as exemplified by the top 30 rank based on fluorescence Z-score for each pool (*Figure 1I*). The potential targets include the housekeeping genes ARIH1, ARHGEF39, and DNAJC7, as well as the poorly described C6orf165 (CFAP206). We also observed reactivity for the known autoantigens SCL-70 and Sm/RNP, and the single-stranded binding protein SSBP1.

Thus, using single-cell RNA-seq, we confirmed that in the 564Igi mixed BM chimeric autoimmune model, B cells with a WT BCR repertoire break tolerance, expand in spontaneous GCs and develop into MemBs and ASCs in a seemingly unrestricted manner, much like they do in response to foreign antigens.

## ASCs have alternative terminal states

As ASCs progress in development, they downregulate *Cd19* and upregulate *Sdc1* (Syndecan-1, CD138). Indeed, clear changes in the expression level of these markers were observed across the seven ASC subclusters (*Figure 2A, B*, left), and this information was used to define cluster ASC_Early_1, which bears the highest *Cd19* and lowest *Sdc1* expression, as the start of pseudotime analysis with Slingshot (*Street et al., 2018*). Two pseudotime lineages were observed (*Figure 2B*, right), concluding in two different clusters: ASC_Late_1 and ASC_Late_2. While the seven clusters displayed a similar overall transcriptional profile (*Figure 2—figure supplement 1A*), they could be characterized by the expression of marker genes, among them: *Fcmr*, *Cd52*, *Il21r*, *Cd74*, and *Ms4a1* for ASC_early1; *Vim*, *Mki67*, *Ccna2*, *Ccnb2*, and *Top2a* for ASC_early_2; *Ssr3*, *Selenof*, *Selenok*, *Lgals1*, and *Kdelr2* for ASC_Mid_1; *Slpi*, *Kdelr1*, *Bst2*, *Eif5a*, *Ctsb*, and *Ubb* for ASC_Mid_2; *Cxcr4*, *Ccnd2*, *Lars2*, *Itgal*, and *Trp53inp1* for ASC_Mid_3; *Ptprc*, *Atf6*, *Hspa5*, *Slamf7*, and *Itga4* for ASC_Late_1; *Bcl2*, *Ly6e*, *Il2rg*, *Tnfrsf17*, *Tnfrsf13b*, and *Lamp1* for ASC_Late_2 (*Figure 2—figure supplement 1B*). All seven subclusters were observed both in autoimmune and immunized chimeras (*Figure 2—figure supplement 1C*).

The two terminal clusters differed in key aspects. *Xbp1* (X-box binding protein 1), a key modulator of the endoplasmic reticulum (ER) stress response and the unfolded protein response and a major

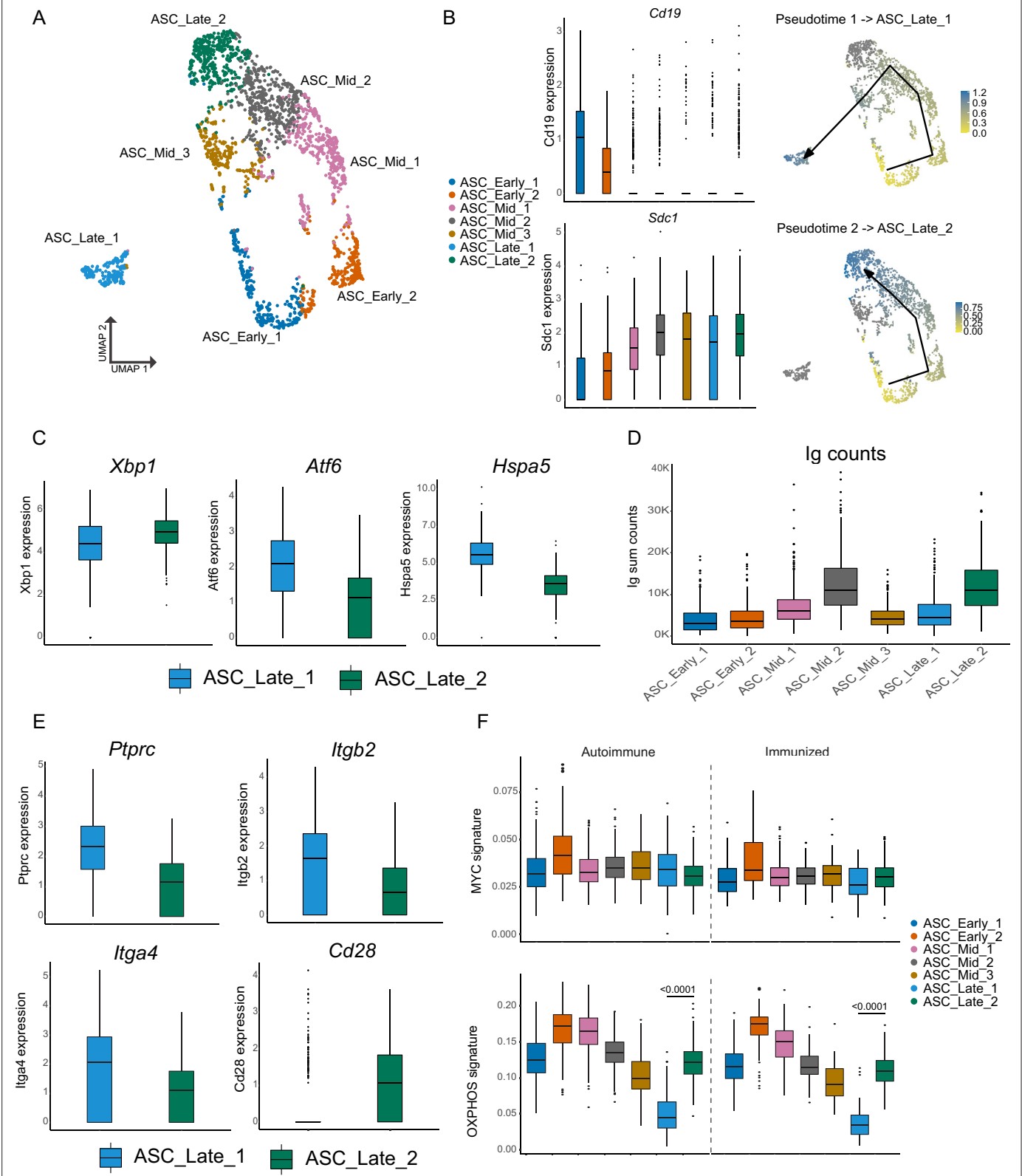

**Figure 2.** Antibody-secreting cells (ASCs) mature into two distinct terminal clusters. (**A**) Two-dimensional UMAP representation of ASCs and their assigned subcluster for all ASC cells from autoimmune and immunized chimeras. (**B**) Cluster normalized gene expression levels for *Cd19* and *Sdc1* (left) and Slingshot pseudotime-based trajectory of ASCs (right). Tukey boxplot for gene expression. (**C**) Normalized gene expression level of *Xbp1*, *Atf6*, and *Hspa5* in ASC_Late_1 and ASC_Late_2. (**D**) Total counts of Ig transcripts for all ASC subclusters, including both autoimmune and immunized chimeras.

*Figure 2 continued on next page*

*Figure 2 continued*

(**E**) Normalized gene expression level of surface markers *Ptprc, Itgb2, Itga4*, and *Cd28* in ASC_Late_1 and ASC_Late_2. (**F**) Single-cell level scoring for MYC and OXPHOS signature profiling of ASC subclusters with AUCell, split by condition. Statistical values correspond to one-way analysis of variance (ANOVA) with Tukey correction for multiple comparisons (**F**).

The online version of this article includes the following source data and figure supplement(s) for figure 2:

**Figure supplement 1.** Complexity of the antibody-secreting cell (ASC) compartment.

**Source data 1.** Data for *Figure 2F*, antibody-secreting cell (ASC) OXPHOS autoimmune graph.

**Source data 2.** Data for *Figure 2F*, antibody-secreting cell (ASC) OXPHOS immunized graph.

regulator of the transition from B cell to ASC, had slightly more expression in ASC_Late_2 than in ASC_Late_1, whereas ASC_Late_1 had *Atf6* and *Hspa5* as markers (*Figure 2C*). These two genes encode Activation transcription factor 6 and Heat Shock Protein Family A (HSP70) Member 5, which are also involved in the regulation of ER stress. These results indicate that terminally differentiated ASCs might use different mechanisms to handle ER stress. Moreover, ASC_Late_2 had more reads mapped to Ig genes than ASC_Late_1 (*Figure 2D*). Similar observations regarding differential usage of ER stress modulation pathways have been made using LPS-stimulated B cells (*Scharer et al., 2020*). However, we are not aware of other reports identifying in vivo genetic and repertoire differences of these terminally differentiated ASCs in spontaneous GCs. Although few surface markers were unique to the two subclusters, higher expression of *Ptprc* (CD45, B220), *Itga4* (CD49d), and *Itgb2* (CD18) were observed in ASC_Late_1 cells and *Cd28* in ASC_Late_2 (*Figure 2E*).

Further characterization of the subclusters using gene set activity analysis with AUCell (*Aibar et al., 2017*) identified similar levels of gene expression for the Myc pathway, as expected, with the exception of ASC_early_2, which was characterized by proliferation markers. Importantly, a higher score signature of oxidative phosphorylation (OXPHOS) in ASC_Late_2 relative to ASC_Late_1 was observed, suggesting specific metabolic requirements (*Figure 2F*).

We analyzed the antibody repertoire for the two terminal clusters, taking advantage of the paired Ig heavy and light chain sequence results. In our dataset, ASC_Late_2 consisted mainly of IgM ASCs, regardless of the immune status (*Figure 3A*), whereas ASC_Late_1 cells had a highly diverse isotype usage, with IgG2c being the most common in autoimmune mice. ASC_Late_1 cells accumulated more replacement mutations than ASC_Late_2 cells, and more replacement mutations were observed in immunized than in autoimmune chimeras (*Figure 3B*). Comparison of the clonal composition of the ASC clusters revealed extensive clonal overlap (*Figure 3C*) in both conditions. This indicates that although the terminal states have transcriptomic and repertoire differences, clones have members with a potential of expansion in any cluster. Moreover, analysis of clonal usage of V genes identified multiple instances of redundancy in the ASC_Late clusters, for example IGHV 1–26, 1–69, and 14–4, among others (*Figure 3D*).

In summary, we identified multiple ASC clusters as they differentiated in both autoimmune and immunized environments, with pseudotime analysis suggesting two terminal states with divergent transcriptomic and VDJ maturation profiles, as well as potentially distinct capacities for antibody secretion but that nonetheless share or contain similar clonal members.

## MemBs are composed of diverse clusters with differences in their transcriptome and repertoire

An important factor in the pathology of autoimmune disease is the self-sustaining chronicity of auto-reactive B cells, which suggests the presence of MemBs. However, it is unclear whether the MemB compartment is similar in autoimmune and immunized responses, and little is known about its internal complexity. To address this question, we reclustered B cells identified as MemBs (*Figure 1B*), identifying four subclusters (*Figure 4A*). Notably, all MemB subclusters were observed for both autoimmune and immunized mice (*Figure 4B*). Remarkably, clusters MemB_1 and MemB_3 accumulated more replacement mutations than clusters MemB_2 and MemB_4, though the latter two still contained cells with similar maximum mutations as MemB_1 and MemB_3 (*Figure 4C*). Examination of Ig isotype revealed that Cluster MemB_2 was mostly composed of IgM cells in both autoimmune and immunized chimeras. A hallmark of the inflammatory interferon-driven response common in lupus and viral infections is expression of IgG2c. While this isotype was nearly absent among the four subclusters of

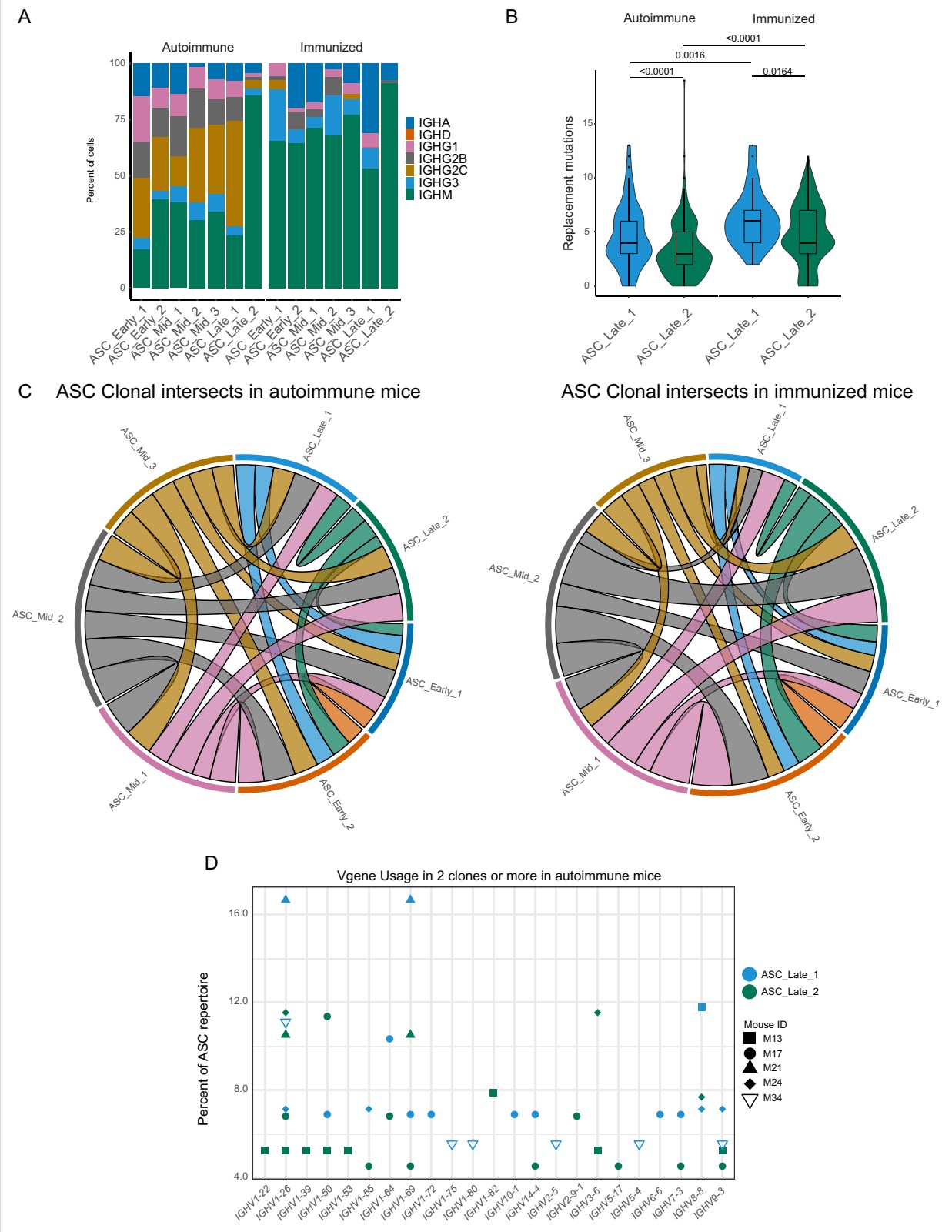

**Figure 3.** Terminal states of antibody-secreting cells (ASCs) have different repertoire characteristics but maintain clonal overlap. (**A**) ASC subcluster isotype usage proportion by condition. (**B**) Replacement mutation accumulation in ASC_Late_1 and ASC_Late_2 by condition. Violin plot with Tukey boxplot overlay. (**C**) ASC subcluster clonal intersects in autoimmune (left) and immunized (right) chimeras, including all ASCs per condition. Connector ribbons' width represents number of shared clones (see ***Supplementary file 1***). (**D**) Clonal V-gene usage in ASC_Late_1 and ASC_Late_2 of

*Figure 3 continued on next page*

*Figure 3 continued*

autoimmune chimeras, per mouse. Only V genes used in two clones or more are displayed. Statistical values correspond to two-tailed Mann–Whitney test (**B**).

The online version of this article includes the following source data for figure 3:

**Source data 1.** Data for *Figure 3B*.

MemB in the immunized mice, all four subclusters of MemB in autoimmune chimeras included the inflammatory isotype (*Figure 4D*).

Overall, although distinct, the clusters displayed extensive overlap in gene expression (*Figure 4—figure supplement 1A*). Comparing the transcriptomic profiles of the four clusters revealed distinct markers for each cluster (*Figure 4E*, *Figure 4—figure supplement 1B*). MemB_1 cells had the highest expression of *Cd83*, a common B cell activation marker typically observed in LZ B cells, and of *Il4i1*, plus they had higher expression of *Apex1*, *Eif5a*, *Eif4a1*, *Slc25a5*, *C1qbp*, and *Mif*. MemB_2 cells had the highest expression of *Fcrl5* and *Cd72*, similar to DN2 cells in humans (*Jenks et al., 2018*) and atypical memory B cells in mice and humans (*Kim et al., 2019*), along with higher expression of *Zeb2*, *Apoe*, *Cd38*, *Cd81*, *Itgb1*, and *Syk*, among others. Markers of MemB_3 were *S100a10*, *Vim*, *Lgals1*, *Ass1*, *Itgb7*, *Sec61b*, *Anxa2*, and *Stk38*. Interestingly, Vimentin (*Vim*) is a cytoskeleton component important for the filament reorganization following BCR stimulation (*Tsui et al., 2018*). Although unique markers for MemB_4 were scarce, *Fcer2a* (encoding for CD23) and *Icosl* showed the highest expression. Other MemB_4 markers include *Cd55*, *Il2rg*, *Ets1*, *Ltb*, *Lmo2*, and *Zfp36*.

Altogether, our interpretation is that MemB_1 likely represents recent GC-derived B cells, as it shows the highest single-cell score for the MYC pathway (*Figure 4—figure supplement 1C*), MemB_2 corresponds to the AtMemBs described in chronic antigen exposure and aging and is analogous to DN2 cells in humans, MemB_3 corresponds to MemBs in an activated state, as it had the highest score for a signature of genes downregulated by Pten, a major regulator of B cell activation (*Figure 4—figure supplement 1D*), and MemB_4 corresponds to a memory compartment dependent on interacting with T cells for reactivation.

To further understand the relationship between MemB subclusters, we investigated the clones present in the four MemB clusters. Remarkably, we found that in both autoimmune and immunized mice (*Figure 5A*), all MemB clusters share clones with one another. This suggests that, regardless of their repertoire origin and specificities, MemBs can acquire any of the different cluster characteristics and that circulation among them might exist. Moreover, as observed in ASCs, regardless of their transcriptomic and repertoire differences, subclusters can have similar V gene usage for IGHV 1–15, 1–42, 1–53, 1–64, 1–75, 3–6, and 9–3, among others (*Figure 5B*).

To evaluate the relationship between potential reservoir (memory) and active response (secretion), we looked for clonal intersection between ASCs and MemBs. We found that clones across all the MemB clusters can seed all the different ASC clusters, regardless of the autoimmune or immunized context (*Figure 5C*). Cluster MemB_4 showed higher clonal contribution in immunized chimeras than in autoimmune chimeras, and in both conditions, most clones were shared between MemB_3 and all the ASC subclusters.

## Validation of the GC origin of the MemB compartment

We generated a new set of autoimmune chimeras (n = 5, *Figure 6A*), which were maintained on a tamoxifen diet for a period of 8 weeks before removing tamoxifen for 4 weeks before analysis. Among the MemBs (AID-EYFP+ GL7− CD138−), we found that MemB_2 (FCRL5+) and MemB_4 (CD23+) cells were in similar proportions in all mice (*Figure 6B*, *Figure 6—figure supplement 1A*).

Recent studies show that *Aicda* (AID) expression can precede the commitment and formation of GC B cells at least in immunized mice (*Roco et al., 2019*). This could suggest that the observations with AID reporter mice could also include non-GC-derived MemBs. Indeed, formation of extrafollicular MemBs cannot be ruled out from the AID-based model (*Lee et al., 2011*; *Toyama et al., 2002*). As an alternative solution for GC fate mapping, we used *S1pr2*, which is an established fate marker for GC-derived B cells (*Shinnakasu et al., 2016*). To confirm the observations made with our single-cell dataset and Aicda-CreERT2-EYFP reporter chimeras, a new set of chimeras using the S1pr2-CreERT2-tdTomato reporter system in combination with 564Igi BM were constructed (n = 3, *Figure 6C*). The

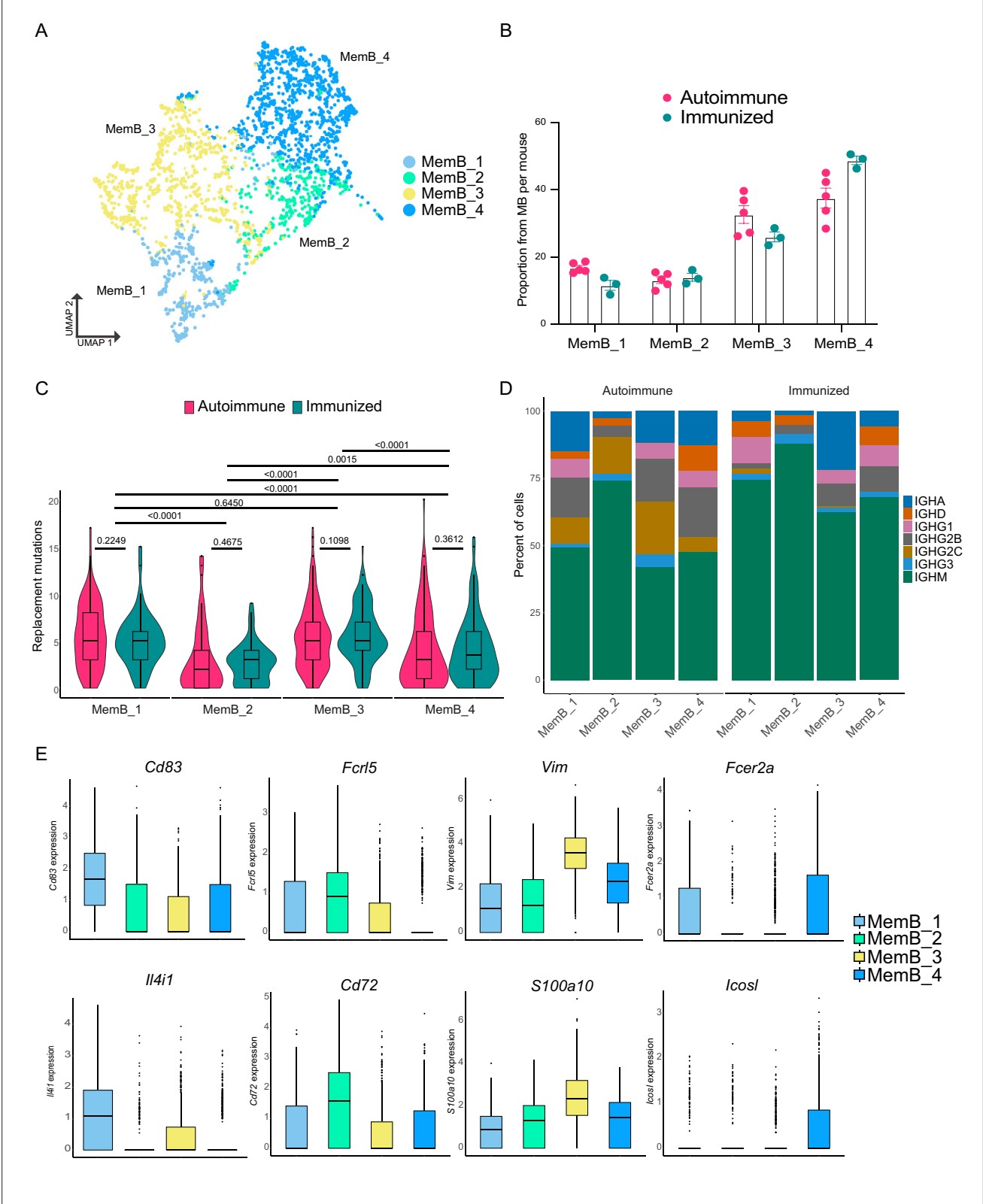

**Figure 4.** Memory B cells (MemBs) are grouped in four subclusters with distinct transcriptomic and BCR profiles. (**A**) UMAP distribution of all cells in the MemB compartment colored by subclusters. (**B**) Proportion of MemBs in each subcluster per mouse. Each dot represents a mouse. (**C**) Replacement mutation accumulation in each MemB subcluster by condition. Violin plots embedded with Tukey boxplots. (**D**) Proportion of isotype usage for each

*Figure 4 continued on next page*

*Figure 4 continued*

MemB subcluster by condition. (**E**) Normalized gene expression of markers that characterize each MemB subcluster. Combined expression from autoimmune and immunized chimeras. Statistical values correspond to two-tailed Mann–Whitney tests (**C**).

The online version of this article includes the following source data and figure supplement(s) for figure 4:

**Figure supplement 1.** Complexity of the memory B cell (MemB) compartment.

**Source data 1.** Data for *Figure 4B*.

**Source data 2.** Data for *Figure 4C*, by cluster.

**Source data 3.** Data for *Figure 4C*, within cluster.

donor *S1pr2* reporter mice were also crossed with Prdm1-EYFP reporter mice to distinguish GC-derived ASCs.

As predicted, MemB cells (S1pr2tdTomato+ Prdm1EYFP−) were observed among the GL7− CD138− population based on flow cytometry. Further analysis of S1pr2tdTomato+ B cells identified CD23+, FCRL5+ MemBs, corroborating our previous observations (*Figure 6D*, *Figure 6—figure supplement 1B*).

To gain insight into the spatial distribution of Memory S1pr2tdTomato+ B cells, cryosections of splenic tissue were characterized by fluorescent scanning confocal microscopy. As expected, S1pr2tdTomato+ cells localized mostly to GCs in the B cell follicles. Interestingly, a substantial number of GC-derived FCRL5+ MemBs localized close to the MZ and particularly near the bridging channels (*Figure 6E*), delineated by CD169+ macrophages.

To confirm the MemB FCRL5+ localization, we injected anti-CD45 antibodies intravenously (i.v.) into autoimmune chimeras (S1pr2-CreERT2-tdTomato:564Igi) 5 min before sacrifice (*Figure 6F*). This labels cells exposed to blood circulation in the spleen, as those residing in the MZ would be (*Cinamon et al., 2008*; *Song et al., 2022*). We observed a significant difference in the i.v. labeling for MemB FCRL5+ in contrast to MemB CD23+ cells, confirming a preferential MZ localization (*Figure 6G*, *Figure 6—figure supplement 1C*). We confirmed the efficiency of MZ preferential labeling using a traditional gating strategy for MZ and follicular B cells in the same mice (*Figure 6—figure supplement 1D*). This privileged location would allow MemB FCRL5+ cells to quickly reactivate upon exposure to foreign antigen but also perpetuate a detrimental response when reacting to self-antigens.

Altogether, we found an underlying complexity of MemBs that, in autoimmune and immunized mice, can be further subdivided into groups that share clonality with ASCs and have transcriptomic and VDJ repertoire signatures suggesting distinct roles in the immune response. Indeed, Memb_2 cells are evidence of memory contribution to the MZ and that the GC-derived MemB compartment can contain DN2-like cells and have the capacity to contribute to pathology through the preservation of harmful self-reactive specificities.

## Discussion

The study of antibody-driven autoimmunity often relies on the use of mice with knock-in BCRs specific for a self-antigen. This approach has been important in establishing many of the current concepts of B cell autoimmunity such as receptor editing, follicular exclusion, and clonal deletion. However, introducing a BCR with a certain pre-determined affinity can potentially bias the observations of the rules governing the activation of those cells. Moreover, conclusions drawn from monogenic BCR mice are unlikely to accurately predict the response to self-antigen when a full repertoire of cells are competing for self-antigen and T cell help. Self-reactivity exists as a spectrum of affinities controlled by sensitive mechanisms that might have different regulatory properties (*Smith et al., 2019*; *Tan et al., 2019*).

The mixed BM chimera model used in our study allows for the pre-defined self-reactive cells to kick-start autoimmunity. In this environment, naive self-reactive WT B cells that normally undergo negative selection can spontaneously become activated and enter self-reactive GCs. Although this approach still depends on a BCR knock-in transgene, it provides a model to study the development of WT B cells as they mature in an autoimmune environment (*Degn et al., 2017*).

We found that self-reactive WT B cells mature and reach pausi-clonality through GC selection and eventually enter all the major mature B cell compartments, similar to that of the immunized controls. Both differential and overlapping clonal usage of V genes were observed in the autoimmune and

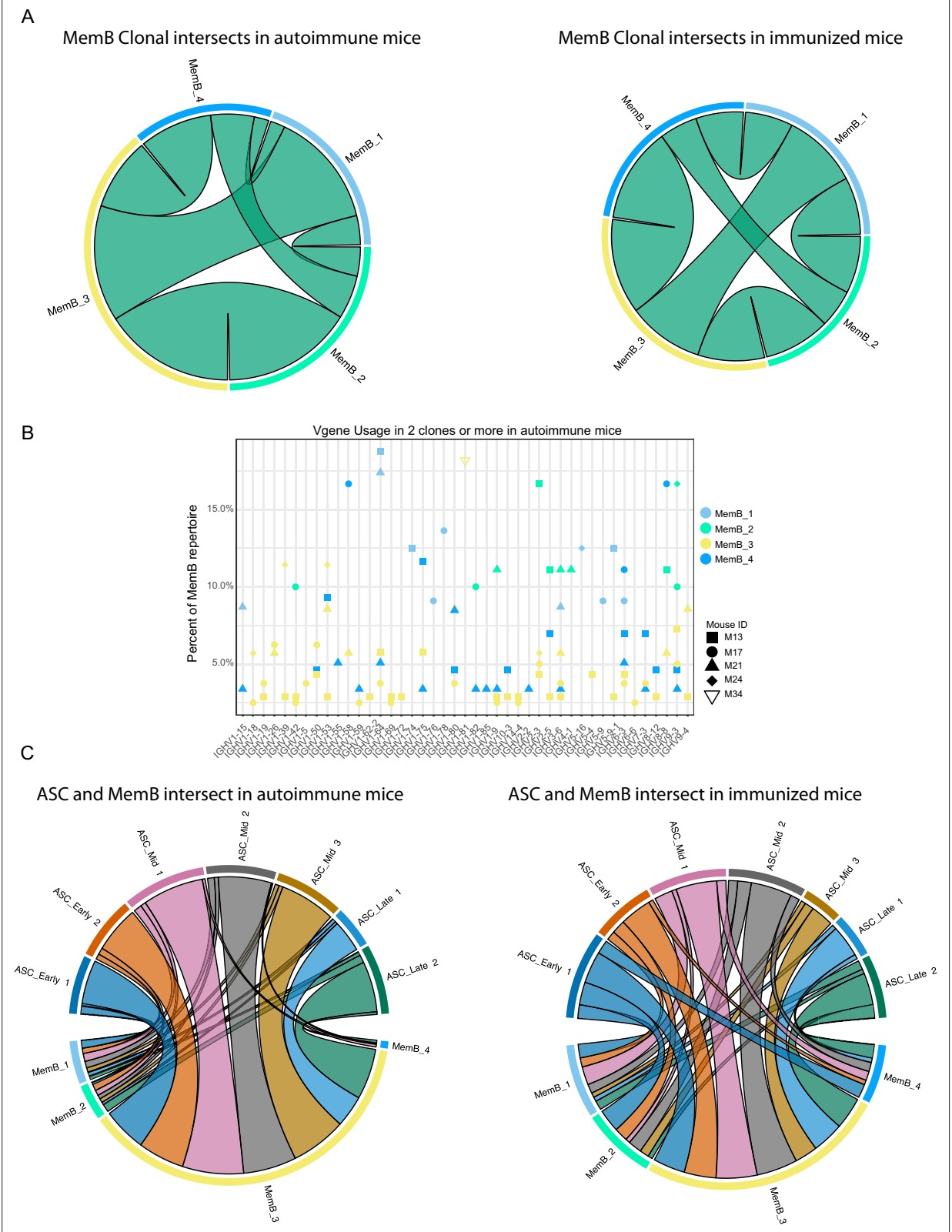

**Figure 5.** Clonal relations among and between memory B cell (MemB) and antibody-secreting cell (ASC) subclusters. (**A**) Clonal intersects among MemB subclusters in autoimmune (left) and immunized (right) chimeras. Connector ribbons' width represents number of shared clones (see *Supplementary file 1*). (**B**) Clonal V gene usage in MemB subclusters for the autoimmune chimeras, per mouse. Only V genes used in two clones or more. (**C**) Clonal intersects between ASC and MemB subclusters in autoimmune (left) and immunized (right) chimeras. Connector ribbons' width represents number of shared clones (see *Supplementary file 1*).

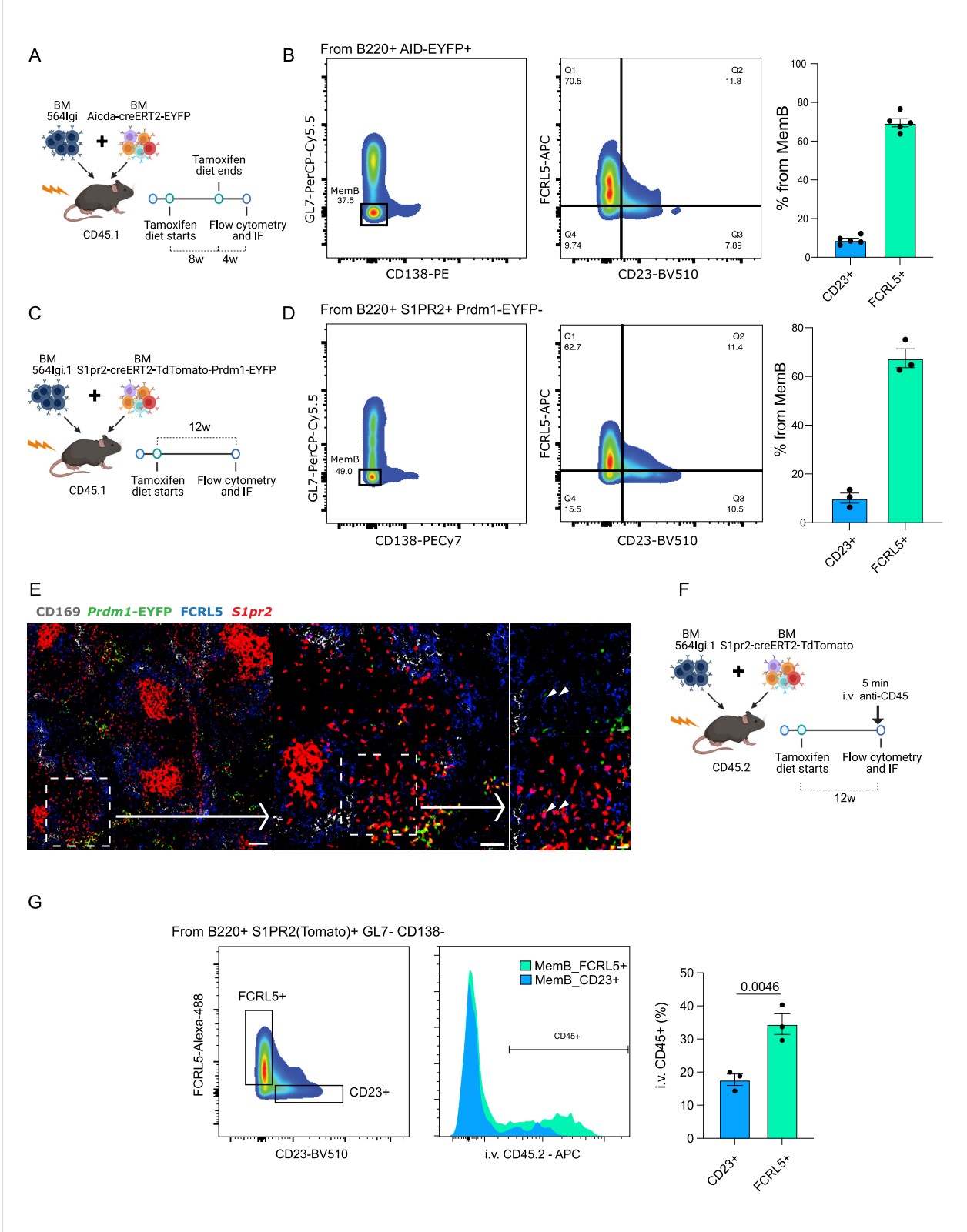

**Figure 6.** FCRL5+ and CD23+ fate-mapped memory B (MemB) cells have different preferential localization. (**A**) Experimental design to validate presence of fate-mapped FCRL5+ and CD23+ MemBs. (**B**) Identification of discrete populations of FCRL5+ and CD23+ MemB cells with Aicda-CreERT2-EYFP reporter mice by flow cytometry (n = 5 chimeras). (**C**) Experimental design for using S1pr2-CreERT2-tdTomato fate mapping in autoimmune chimeras. (**D**) Identification of discrete populations of FCRL5+ and CD23+ MemB cells with S1pr2-CreERT2-tdTomato fate mapping in autoimmune

*Figure 6 continued on next page*

*Figure 6 continued*

chimeras by flow cytometry (*n* = 3 chimeras). (**E**) Splenic localization of FCRL5+ S1PR2(Tomato)+ MemB cells by confocal microscopy. Overview of a spleen from autoimmune chimeras (**Figure 3C**), left, and selected area, center, delineates a bridging channel for a close-up examination of FCRL5+ S1pr2tomato+, right. Arrow points to FCRL5+ MemBs. Scale bars represent 100 µm (left), 50 µm (center), and 20 µm (right). Arrow heads point to FCRL5+ S1PR2(Tomato)+ cells. (**F**) Experimental design for in vivo marginal zone labeling. S1pr2-CreERT2-tdTomato:564Igi chimeras intravenously (i.v.) injected with 5 µg of anti-CD45-APC for 5 min before organ extraction. (**G**) Flow cytometry gate strategy to evaluate i.v. CD45 labeling between MemB FCRL5+ and MemB CD23+ cells. Statistical values correspond to one-tailed unpaired Student *t*-test (**G**).

The online version of this article includes the following figure supplement(s) for figure 6:

**Figure supplement 1.** Controls for validation of memory B cell (MemB) clusters by flow cytometry.

immunized chimeras. For example, among the most used V genes in both conditions were IGHV 1–53. This VH family member appears in many scenarios such as MZ B cells reacting to HIV gp120 immunization (*Pujanauski et al., 2013*) and self-reactive clones in FcgRIIB-deficient mice (*Tiller et al., 2010*; *van der Poel et al., 2019*). Intriguingly, 1–53 is among the most commonly used genes in the mouse response to the hapten group NP (*Xue et al., 2019*), most likely due to its similarity to IGHV 1–72. Thus, IGHV 1–53 can be considered part of the collection of baseline repertoire that gives rise to 'polyreactive' antibodies. Although polyreactive antibodies may be detrimental in autoimmunity, they are likely retained in the repertoire for use when a quick response to a foreign antigen is needed. Notably, mouse IGHV 1–53 is the mouse ortholog to human IGHV 1–69 (*Ighv1-53, 2023*). IGHV 1–69 is commonly observed in broadly neutralizing antibodies in the anti-viral response, but together with IGHV 4–34, it is also a feature of antibodies in autoimmune disease (*Bashford-Rogers et al., 2018*; *Watson and Breden, 2012*). Additionally, the ortholog to IGHV 4–34 in mice is IGHV 3–6 (*Ighv3-6, 2023*), which was the most clonally used gene in our autoimmune chimeras. This suggests that self-reactivity can be driven by predominant V genes that might initially be polyreactive but are further tailored in GCs. Altogether, these data suggest that there might be different ways to generate self-reactive pathogenic clones, starting with common V genes but leading to the usage of more specific ones. Distinguishing the contributions of originally polyreactive vs non-polyreactive clones will allow for better understanding of the mechanisms that lead to epitope spreading.

Many characteristics have been attributed to ASCs: short- vs long-lived, plasmablasts vs plasma cells, and EF- vs GC-derived. We observed two terminal clusters of ASC development with clear distinctions in the number of Ig transcripts, metabolic requirements and ER-stress-related gene expression. ASC_Late_1 had lower Ig counts and score signature for oxidative phosphorylation, together suggesting a reduced rate of antibody secretion (*Lam et al., 2018a*). Intriguingly, ASC_Late_1 had more replacement mutations than ASC_Late_2 and a more diversified usage of isotype. Given that MemBs have accumulated mutations over-time in the GC and that our sequencing dataset was processed ~5 weeks after tamoxifen treatment while the lifespan of short-lived ASCs is estimated at ~2 weeks, reactivated MemBs are the likely origin of most ASC_Late_1 cells. Although pseudotime analysis leads us to classify ASC_Late_2 as one of the two terminal clusters, the immediately preceding cluster ASC_Mid_2 also possesses characteristics of mature ASCs as well, including prominent expression of *Cxcr4* and *Slpi*, which is a marker for BM and long-lived ASCs (*Lam and Bhattacharya, 2018b*), suggesting it could represent another terminal state.

We identified four MemB subclusters. MemB_1 has characteristics of GC B cells and likely represents recently egressed cells. MemB_2 has similar markers as atMemB and DN2 cells. MemB_3 shares similar characteristics to recently activated B cells. MemB_4 shows patterns of interaction with T cells (and resembles DN4 in humans). We suspect that the different properties of MemB_2 and MemB_4 exist to allow the B cell compartment to respond adequately in a diversity of contexts. Indeed, we observed distribution of MemB_2 GC-derived cells in the MZ and in bridging channels – convenient locations for immediate antigen surveillance and response. MemB_4 might represent T-dependent bona fide long-lived MemBs, maintaining a record of previous exposure and requiring involvement of T cells for retriggering.

Finally, we found that, to varying degrees, all subclusters of ASCs and MemBs shared members of the same clones. MemB_3 showed striking predominance in clonal overlap with ASCs of all subclusters, indicating this cluster is an activated version of all the other MemB subpopulations. The clonal overlap between all these subclusters was surprising, although not completely unexpected after finding overlapping clonality among ASCs and MemBs when analyzed separately. This overlap likely

provides a wider safety-net, an insurance of long-lived memory and a readiness to reactivate to deal with prospective pathogens. On the other hand, this implicates that the efforts to target any one cellular compartment for the sake of therapeutically ablating autoimmunity may overlook the presence of shared self-antigen specificity across different cellular niches.

Overall, using a WT BCR repertoire mouse model that maps spontaneous GC-derived cells, we observed a diversity of subgroups, reflecting the inner complexities of the ASC and MemB compartments, with specific transcriptomic and repertoire characteristics, but with underlying redundant clonality.

# Materials and methods

**Key resources table**

| Reagent type (species) or resource | Designation | Source or reference | Identifiers | Additional information |
|---|---|---|---|---|
| Strain, strain background (*M. musculus*) | *Aicda*CreERT2 | C-A Reynaud, J-C Weill. (Institut Necker) | | |
| Strain, strain background (*M. musculus*) | *S1pr2*CreERT2 BAC-transgenic | T. Kurosaki (RIKEN-Yokohama) | | |
| Strain, strain background (*M. musculus*) | B6.Cg-Tg(Prdm1-EYFP)1Mnz/J | Jackson Laboratories | | |
| Strain, strain background (*M. musculus*) | C57BL/6J | Jackson Laboratories | | |
| Strain, strain background (*M. musculus*) | B6.SJL | Jackson Laboratories | | |
| Strain, strain background (*M. musculus*) | 564Igi | Theresa Imanishi-Kari (Tufts University) | | |
| Strain, strain background (*M. musculus*) | 564Igi.1 | This manuscript | | |
| Antibody | anti-CD45.2-APC (104) (mouse monoclonal) | Biolegend | 109814 | 1:300 |
| Antibody | anti-CD23-BV510 (B3B4) (rat monoclonal) | Biolegend | 101623 | 1:300 |
| Antibody | anti-B220-PacBlue (RA3-6B2) (rat monoclonal) | Biolegend | 103227 | 1:300 |
| Antibody | anti-B220-PerCP/Cy5.5 (RA3-6B2) (rat monoclonal) | Biolegend | 103234 | 1:300 |
| Antibody | anti-GL7-PacBlue (GL7)(rat monoclonal) | Biolegend | 144614 | 1:300 |
| Antibody | anti-GL7-PerCP/Cy5.5 (GL7) (rat monoclonal) | Biolegend | 144610 | 1:300 |
| Antibody | anti-CD138-Biotin (281-2) (rat monoclonal) | Biolegend | 142512 | 1:300 |
| Antibody | anti-CD138-PE (281-2)(rat monoclonal) | Biolegend | 142504 | 1:300 |
| Antibody | anti-CD45.1-APC (A20)(mouse monoclonal) | Biolegend | 110714 | 1:300 |
| Peptide, recombinant protein | Streptavidin | Biolegend | 405206 | |
| Antibody | anti-FCRL5-Alexa488 (sheep polyclonal) | biotechne | FAB6757G | 1:10 |
| Antibody | anti-FCRL5-APC (sheep polyclonal) | biotechne | FAB6757A | 1:10 |
| Commercial assay, kit | Fixable Viability Dye eFluor 780 | Thermo Fisher | 65-0865-14 | 1:1000 |
| Commercial assay, kit | Pan B Cell Isolation Kit II, mouse | Miltenyi | 130-104-443 | |
| Peptide, recombinant protein | NP-OVA | Biosearch | N-5051-10 | |
| Chemical compound, drug | Tamoxifen | Sigma-Aldrich | T5648-5G | |
| Chemical compound, drug | Imject Alum | Thermo Fisher Scientific | 77161 | |

*Continued on next page*

*Continued*

| Reagent type (species) or resource | Designation | Source or reference | Identifiers | Additional information |
|---|---|---|---|---|
| Software, algorithm | FlowJo | FlowJo LLC | | |
| Software, algorithm | Prism | GraphPad | | |
| Software, algorithm | R | R Foundation | | 4.1.2 |
| Software, algorithm | Cellranger | 10× Genomics | | 5.0.1 |
| Software, algorithm | GC tree | https://github.com/matsengrp/gctree (*DeWitt et al., 2018*) | | |
| Commercial assay, kit | HuProt v4.0 array | CDI Labs | | |
| Commercial assay, kit | Single Cell Immune Profiling | 10× Genomics | | |
| Software, algorithm | Immcantation | https://immcantation.readthedocs.io/en/stable/ | | |

## Study design

The purpose of this study was to characterize the post-GC populations of antibody-secreting and MemBs in the context of autoimmunity. We used GC fate-mapping and single-cell transcriptomics coupled to BCR repertoire analysis. The BM chimeric mouse model we used allows to track WT B cells as they develop and exit from spontaneous GCs. We contrasted the autoimmune chimeras with NP-OVA immunized chimeras.

## Mice

C57BL/6J and B6.SJL (CD45.1), B6.Cg-Tg(*Prdm1*-EYFP)1Mnz/J (Blimp-EYFP) were obtained from Jackson Laboratories. *Aicda*-CreERT2 flox-stop-flox- EYFP mice (*Dogan et al., 2009*) were from Claude-Agnes Reynaud and Jean-Claude Weill (Institut Necker). *S1pr2*-CreERT2 BAC-transgenic mice (*Shinnakasu et al., 2016*) were generated and generously provided by T. Kurosaki (RIKEN-Yokohama). 564Igi mice (*Berland et al., 2006*) were originally provided by Theresa Imanishi-Kari (Tufts University) and were maintained in-house. 564.1 mice were generated by crossing 564Igi and B6.SJL (CD45.1) mice. 564Igi and 564.1Igi mice were genotyped by ddPCR using the primers: 564Igi_H_Fwd-cacagattcttagtttttcaa, 564Igi_H_Rev-tggagctatatcatcctcttt, 564Igi_K_Fwd-ccagtgcagattttcagcttc, 564Igi_K_Rev-cagcttggtcccagcaccgaa, mRPP30_Fwd-tgaccctatcagaggactgc, and mRPP30_Rev-ctctgcaatttgtggacacg. All mice were bred and maintained in the AAALAC-accredited facility at Harvard Medical School. Mice were specific pathogen-free and maintained under a 12-hr light/dark cycle with standard chow diet. Both male and female mice were used. All animal experiments were conducted in accordance with the guidelines of the Laboratory Animal Center of National Institutes of Health. The Institutional Animal Care and Use Committee of Harvard Medical School approved all animal protocols (protocol number IS111).

## Immunization and antibody injection

For single-cell sequencing, non-autoimmune chimeric mice were i.p. immunized with 100 µg of 4-hydroxy-3-nitrophenylacetyl hapten conjugated to ovalbumin (NP-OVA, Biosearch) in 50 µl Hanks' Balanced Salt Solution (HBSS) precipitated in 50 µl of Imject Alum (Thermo Scientific) 6–8 weeks after irradiation and BM reconstitution. Four weeks after immunization, mice received an intraperitoneal booster immunization of 100 µg of NP-OVA in 100 µl HBSS. Mice were sacrificed 2 weeks after boost.

For in vivo labeling of MZ proximal cells, 5 min prior to euthanasia each mouse was injected retro-orbitally with 5 µg of anti-CD45.2-APC (Biolegend) diluted in 200 µl of phosphate-buffered saline (PBS).

## Tissue processing

Mice were euthanized by cervical dislocation under isoflurane induced anesthesia. Spleens were extracted, dissected, and processed for immunofluorescence microscopy or flow cytometry analysis and cell sorting.

## BM chimeras

BM chimeras were prepared using marrow from 564Igi mice and BCR WT donor as previously described (*Degn et al., 2017*). Mice were lethally irradiated at 1100 rad and kept on antibiotics

(sulfamethoxazole/trimethoprim) through drinking water for 7 days after irradiation. Femurs and tibia from donor mice were cleaned from muscle tissue and subsequently rinsed with cell transfer buffer (HBSS supplemented with 10 mM N-2-hydroxyethylpiperazine-N-2-ethane sulfonic acid (HEPES), 1 mM Ethylenediaminetetraacetic acid (EDTA), and 2% heat inactivated fetal bovine serum). Marrow was extracted from bones by crushing them using mortar and pestle and the detached cells were resuspended in cell transfer buffer and passed through a 70-mM sterile filter. Cells were counted by using an erythrocyte lysed aliquot. All autoimmune chimeras were prepared at 2:1 ratio for 564Igi:WT-reporter, with WT-reporter and irradiated hosts genotype as specified for each experiment. BM recipients received $15–20 \times 10^6$ cells i.v. in 100 ml cell transfer buffer by retroorbital i.v. injection approximately 8 hr post irradiation.

### Fate-mapping tamoxifen induction

Mice were exposed to tamoxifen in two different ways, as specified in each experimental design figure. Mice were gavaged with 10 mg of tamoxifen (Sigma) dissolved in Corn Oil at 50 mg/ml twice, at days 4 and 7 post primary immunization in the case of immunized chimeras and at the same time for autoimmune chimeras. For validation experiments, BM chimeric mice were maintained in a tamoxifen enriched diet for the specified timeframes (Envigo) (*Song et al., 2022*).

### Flow cytometry

Spleen fragments were harvested into ice-cold Magnetic-activated cell sorting (MACS) buffer (PBS pH 7.2, 0.5% bovine serum albumin (BSA), and 2 mM EDTA) and mechanically dissociated using pestles in 1.5 ml Eppendorf tubes. Samples spun down at $300 \times g$ for 5 min and resuspended in RBC lysis buffer (155 mM $NH_4Cl$, 12 mM $NaHCO_3$, 0.1 mM EDTA). Samples were washed with MACS buffer and spun down at $300 \times g$ for 5 min. Finally, samples were resuspended in MACS buffer and filtered through 70 µm cell strainers (Corning). Samples were added to 96-well round-bottom plates, spun down and resuspended in 50 µl staining mix and stained for 30 min on ice. Cells were washed with 150 µl of MACS buffer twice by spinning down at $300 \times g$ for 5 min. Finally, cells were resuspended in 200 µl of MACS buffer and transferred to 5 ml fluorescence activated cell sorting (FACS) tubes. The following antibodies and proteins were used: anti-CD23-BV510 (B3B4), anti-B220-PacBlue (RA3-6B2), anti-B220-PerCP/Cy5.5 (RA3-6B2), anti-GL7-PacBlue (GL7), anti-GL7-PerCP/Cy5.5 (GL7), anti-CD138-Biotin (281-2), anti-CD138-PE (281-2), and PE/Cy7-Streptavidin from Biolegend, and anti-FCRL5-Alexa488 and anti-FCRL5-APC (Polyclonal) from biotechne. Viability was determined with Fixable live/dead stain Efluor780 from Thermo Fisher Scientific. A standard 3 lasers configuration (405, 488, and 633 nm) FASCSCanto2, with 8-color and 10 parameter analytical capabilities was used for acquisition. Data were analyzed with FlowJo 10.

### Cell sorting for single-cell sequencing

Spleens from autoimmune and immunized chimeras were processed following viability recommendations for droplet-based Chromium single-cell RNA-seq gene expression (10×). The full organs were dissociated in MACS buffer (PBS 1×, 0.5% BSA, 2 nM EDTA) using syringe plungers and 70 µm cell strainers, spun down at 1000 rpm for 5 min, resuspended in RBC lysis buffer and incubated on ice for 5 min. Samples were washed with MACS buffer and spun down 1000 rpm for min. Cells were enriched by negative selection with a Pan II B cell enrichment kit (Miltenyi) according to the provider specifications. During enrichment incubation cells were also stained for flow sorting. The antibodies used for staining were anti-GL7-PacBlue, ant-CD138-PE, anti-B220-PerCP/Cy5.5, anti-CD38-PECy7, and anti-CD45.1-APC from Biolegend. eFluor780 from Thermo Fisher was used for viability. After enrichment and antibody staining cells were resuspended in resuspension buffer (PBS 1×, 0.04% BSA). Reporter cells were sorted based on EYFP expression (Efluor 780- EYFP+ CD45.1+) at two-way purity sort with a FACSARIA II Special Order system (BD Biosciences) with 355, 405, 488, 640, and 592 nm lasers. Sorted cells were spun down and resuspended in resuspension buffer prior to single-cell encapsulation.

### Droplet-based single-cell sequencing and data processing

The sorted cells were encapsulated with barcoded hydrogels using the Chromium system for Single Cell Immune Profiling, that allows coupling of transcriptomic and VDJ information per cell. cDNA

libraries were prepared according to the manufacturer's recommendations. Library quality control and sequencing (NextSeq 500, Illumina) were performed by the HMS Biopolymers Facility. Cellranger (5.0.1) was used to generate the count matrices for gene expression and the VDJ contigs per cell using the multi function for each mouse. Reads were aligned to a custom mm10 reference genome incorporating the transcript sequences for EYFP and Cre Recombinase. R (4.1.2) was used for further processing the Cellranger gene expression count matrix output using the OSCA workflow (https://bioconductor.org/books/release/OSCA/) as template and using the Single Cell Experiment (SCE) format. Scater and Scran were used for data QC. Cells were subset for less than 5% mitochondrial and 40% ribosomal content. Ig genes were excluded from clustering and posterior expression analysis. Correction was done with fastMNN function from batchelor. Clustering was performed using the Leiden algorithm. Markers for each cluster were identified using the scoreMarkers function from Scran. Pre-defined genesets were used for Oxidative Phosphorylation (GO:0006119), Myc upregulated (*Wang et al., 2020*; *Chen et al., 2021*) and PTEN_DN.V1_DN (GSEA) were used for signature analysis with AUCell. Data visualization was done with dittoSeq.

## BCR repertoire analysis

The VDJ output from Cellranger multifunction was further processed following the Immcantation (https://immcantation.readthedocs.io/en/stable/) recommendations for 10× derived single-cell data. Genes were assigned using IgBlast and the IMGT reference sequence database. Clones were set using the DefineClones function from Change-O, with a 0.1691791 threshold defined using the findtThreshold function from Shazam. countClones (Alakazam) and observedMutations (Shazam) functions were used for quantification.

Phylogenetic trees were generated with GCTree (*DeWitt et al., 2018*) using all VDJ sequences from a given clone and rooted in the germline obtained with CreateGermlines function from Change-O.

## Antibody production and protein array profile

VDJ sequences were synthesized (IDT) for sequences from mice M13 and M21 with barcodes: GGGACCTGTAGCTGCC_B06M13, GATGAGGCATCGGGTC_B10M21, TGTTCCGCAATGGACG_B10M21, CGTGTCTCAAACCCAT_B06M13, GCGCAACCAATCTACG_B06M13, and AAATGCCGTACACCGC_B06M13. Geneblocks were cloned into a modified pVRC8400 vector between a tissue plasminogen activation (TPA) signal sequence and the constant domains of the mouse IgG1 CH1-CH3 and CL (*Kuraoka et al., 2016*). Monoclonal IgG1,k antibodies were produced in suspension by transient transfection of 293F cells, using polyethylenimine (Polysciences). The supernatant was harvested 5 days after transfection. Antibody was recovered from culture supernatant after centrifugation at 4200 rpm for 20 min and clearing with 45 µm filters. IgGs were purified using Protein G agarose (Thermo) and dialyzed in PBS. Purified IgGs were concentrated and stored at 4°C.

For target detection, the monoclonal antibodies were pooled in two different samples and processed for reactivity by CDI Labs with a HuProt v4.0 array, containing 21,000 human proteins. Monoclonal antibodies at 1 µg/ml were diluted in a final volume of 3 ml and were probed with the arrays for 2 hr at room temperature (RT). The arrays were washed according to the company protocol and were probed with the secondary antibody (Alexa-647-goat-anti-mouse IgG gamma-specific) under conditions optimized by CDI Labs for signal detection with GenePix software. Data were processed with CDI's proprietary data analysis software (Z-score analysis).

## Immunofluorescence and confocal microscopy

Following 4% Paraformaldehyde (PFA) fixation on ice for 3 hr, spleens were embedded in OCT (Fisher Healthcare), frozen in dry ice and stored at −80°C. Spleens were cut into 10 µm sections, blocked for 1 hr at RT with Blocking buffer (PBS, 0.01% Tween20, 2% BSA, and 5% Fetal Bovine Serum (FBS)) and stained O.N. at 4°C with antibodies diluted in Blocking buffer. Anti-CD169-BV510 (Biolegend) and anti-FCRL5-APC (biotechne) were used for staining. Slides were washed three times in PBS, 0.01% Tween20 and mounted with FluoroGel (Electron Microscopy Sciences) prior to image acquisition. Images were acquired with an OLYMPUS FV3000R resonant scanning confocal microscope equipped with four laser lines (405, 488, 514, and 633 nm), hybrid galvo and fast resonant scanning capabilities, ultra-sensitive GaAsP detectors with full spectral imaging and motorized XYZ stage for tiling. The images were processed in Fiji.

## Statistical analysis

Two-tailed Mann–Whitney, one-way analysis of variance with Tukey correction and one-tailed unpaired Student $t$-tests were performed with Prism 9 (GraphPad).

## Acknowledgements

We thank J Moore of the Flow and Imaging Cytometry Resource at the BCH PCMM, the BCH Cell Function and Imaging Core for technical assistance, and all the members of the Carroll lab for their feedback and support. This work was funded by NIH grants R01AI130307 and R01AR074105 (MCC). EHA-G. was supported by NIH grants T32GM007753, T32AI007529, and F30AI160909. TvdB was supported by the H2020-MSCA-IF-GF project BEAT (No. 796988) and the Academy Ter Meulen Fund (TMB/16/285). Authors declare that they have no competing interests.

## Additional information

### Funding

| Funder | Grant reference number | Author |
| --- | --- | --- |
| National Institutes of Health | R01AI130307 | Michael C Carroll |
| National Institutes of Health | R01AR074105 | Michael C Carroll |
| National Institutes of Health | T32GM007753 | Elliot H Akama-Garren |
| National Institutes of Health | T32AI007529 | Elliot H Akama-Garren |
| National Institutes of Health | F30AI160909 | Elliot H Akama-Garren |
| H2020 Marie Skłodowska-Curie Actions | BEAT (No. 796988) | Theo van den Broek |
| Koninklijke Nederlandse Akademie van Wetenschappen | Academy Ter Meulen Fund TMB/16/285 | Theo van den Broek |

The funders had no role in study design, data collection, and interpretation, or the decision to submit the work for publication.

### Author contributions

Carlos Castrillon, Conceptualization, Data curation, Software, Formal analysis, Validation, Investigation, Visualization, Methodology, Writing – original draft, Writing – review and editing; Lea Simoni, Theo van den Broek, Cees van der Poel, Conceptualization, Investigation, Methodology, Writing – review and editing; Elliot H Akama-Garren, Investigation, Methodology, Writing – review and editing; Minghe Ma, Investigation, Methodology; Michael C Carroll, Conceptualization, Supervision, Funding acquisition, Writing – review and editing

### Author ORCIDs

Carlos Castrillon ⓘ http://orcid.org/0000-0003-2909-7371
Theo van den Broek ⓘ http://orcid.org/0000-0002-2781-5731
Elliot H Akama-Garren ⓘ http://orcid.org/0000-0002-1690-2055
Minghe Ma ⓘ http://orcid.org/0000-0002-4496-6240
Michael C Carroll ⓘ http://orcid.org/0000-0002-3213-4295

### Ethics

All mice were bred and maintained in the AAALAC-accredited facility at Harvard Medical School. Mice were specific pathogen-free (SPF) and maintained under a 12-hr light/dark cycle with standard chow diet. Both male and female mice were used. All animal experiments were conducted in accordance

with the guidelines of the Laboratory Animal Center of National Institutes of Health. The Institutional Animal Care and Use Committee of Harvard Medical School approved all animal protocols (protocol number IS111).

### Decision letter and Author response
Decision letter https://doi.org/10.7554/eLife.81012.sa1
Author response https://doi.org/10.7554/eLife.81012.sa2

---

## Additional files

### Supplementary files
• Supplementary file 1. Clonal overlap between and among antibody-secreting cell (ASC) and memory B cell (MemB) subsets. This file contains the raw values for the circos plot graphs showing clonal intersects between subsets for *Figures 3C and 5A, C*.
• MDAR checklist

### Data availability
The sequencing data presented in this study have been submitted to the Gene Expression Omnibus under accession number GSE203132. Code is available on GitHub (copy archived at *Castrillon, 2023*).

The following dataset was generated:

| Author(s) | Year | Dataset title | Dataset URL | Database and Identifier |
|---|---|---|---|---|
| Castrillon C, Carroll M | 2023 | Transcriptomic diversity and overlapping clonality across subsets of antibody-secreting and memory B cells from spontaneous germinal centers | https://www.ncbi.nlm.nih.gov/geo/query/acc.cgi?acc=GSE203132 | NCBI Gene Expression Omnibus, GSE203132 |

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
