## [Editor Report]

Understanding the heterogeneity of the B cell response induced in autoimmune individuals is important for the development of therapies designed to target the cells underlying disease progression. Here, the authors use a new model of autoimmunity to assess the heterogeneity of the B cell response using scRNA-seq and scBCR-seq and found that B cell responses are similar to those by exogenous protein immunization.

---

## [Decision Letter]

**Decision letter after peer review:**

Thank you for submitting your article "Complex subsets but redundant clonality after B cells egress from spontaneous germinal centers" for consideration by *eLife*. Your article has been reviewed by 2 peer reviewers, and the evaluation has been overseen by a Reviewing Editor and Betty Diamond as the Senior Editor. The reviewers have opted to remain anonymous.

Essential revisions:

Both of the reviewers recognize the potential interest of this manuscript but have raised several concerns.

1) The authors should perform additional experiments to validate whether the memory B cell subsets identified using Fcrl5 and CD23 are spatially distinct. These experiments should include quantification of the imaging data and a revised flow cytometry gating strategy to identify memory B cell subsets. The imaging data will likely require an improvement in the quality of the staining to better identify the Fcrl5+ cells.

2) The authors should add additional discussion regarding how their memory B cell subsets related to subsets that have been previously identified in mice using markers such as CD80 and PDl2.

3) One of the immunization models used in this study relies on a secondary dose of NP-OVA to activate previously labeled GC-derived B cells. This acute system differs from the chronic B cell activation observed in the autoimmune disease model, in which labeled B cells can be constantly selected into continually ongoing and spontaneous GC and differentiate into antibody-secreting cells. The authors should explain in some lines the differences between these two models and why they decide to compare an acute NP-immunization model with a chronic autoimmune disease model?

4) The model of tamoxifen administration used in figures 1 to 5 (two doses of tamoxifen on days 4 and 7 after initial immunization) and then in figure 6 (tamoxifen diet for 12 weeks) are different. This might result in a different labeling pattern for the cells expressing AID or S1pr2-Cre. Have the authors made sure that both tamoxifen administration schedules are comparable in terms of cell labeling output?

5) In Figure 2, the authors identified two terminally differentiated clusters, ASC_Late_1 and ASC_Late_2, which display different transcriptomics. Using differentially expressed markers, can the authors identify these two ASC populations by flow cytometry as performed in Figure 6 for memory B cells?

6) In figure 3A, ASC_Late_1 has a very diverse isotype usage in the auto-immune mice but not in the immunized ones. On the contrary, ASC_Late_2 in both conditions is overwhelmingly dominated by IgM. Are authors convinced that both ASC populations arise from the autoimmune disease? Are the antibodies generated by both ASC subsets specific for auto-antigens?

7) In Figure 3A, isotype usage for each ASC cluster and in both conditions is shown. According to previous pseudo-time analysis, ASC_Late_2, which mainly displays an IgM BCR, is originated from ASC_Early_1 to ASC_Early_2 to ASC_mild_1 to ASC_mild_2 according to the pseudo-time analysis (figure 2B). However, all these progenitors' clusters display a much lower IgM expression, especially in the auto-immune model, indicating that they have already class-switched and wouldn't be able to give rise to IgM. How do the authors reconcile these two findings?

8) In lines 275 to 281, the authors describe and define the different Memory B cell clusters. According to these lines, MemB_1 is defined as recent GC-derived and MemB_3 corresponds to MemB as activated cells. However, in Figure 6, the authors focused on MemB_2 defined as Fcrl5+ and MemB_4 as CD23+, which are defined as atypical Bmem and Bmem depending on T cell for their activation respectively. Why did the authors choose to focus only on these two clusters? Did the authors confirm that MemB_1 and MemB_3 have a GC origin?

*Reviewer #1 (Recommendations for the authors):*

Specific points

1) The scheme in Figure 1A does not include the exact time points in which mice received tamoxifen and are then sacrificed. Could the authors add this information to the figure for clarity?

2) Figure 1D is missing statistical analysis to support the claims of the authors in lines 87-88-113.

3) Figure 1E displays the phylogenetic tree of the most expanded clone in one of the autoimmune chimeras. The authors claim that expanded clones can be found in all 4 clusters, but looking closely at this figure the MemB cluster is scarcely represented. Previous work (Viant et al. 2020) showed that the memory B cell compartment is largely populated by single cell clones rather than expanded clones. Can the authors quantify the clonal expansion of memory B cells in their model (how many single-cell, 2-cell, 3-cell clones?).

4) In figure 1I, the authors show that two monoclonal antibody pools originating from autoimmune chimeras could bind to some auto-antigens from a human recombinant protein array. Why did the authors check the binding to human autoantigens and not mouse autoantigens? Could they be missing some targets that do not cross-react between mouse and humans?

5) In Figure 2, the authors identified two terminally differentiated clusters, ASC_Late_1 and ASC_Late_2, which display different transcriptomics. Using differentially expressed markers, can the authors identify these two ASC populations by flow cytometry as performed in Figure 6 for memory B cells?

6) In figure 3A, ASC_Late_1 has a very diverse isotype usage in the auto-immune mice but not in the immunized ones. On the contrary, ASC_Late_2 in both conditions is overwhelmingly dominated by IgM. Are authors convinced that both ASC populations arise from the autoimmune disease? Are the antibodies generated by both ASC subsets specific for auto-antigens?

7) In Figure 3A, isotype usage for each ASC cluster and in both conditions is shown. According to previous pseudo time analysis, ASC_Late_2, which mainly displays an IgM BCR, is originated from ASC_Early_1 to ASC_Early_2 to ASC_mild_1 to ASC_mild_2 according to the pseudo-time analysis (figure 2B). However, all these progenitors' clusters display a much lower IgM expression, especially in the auto-immune model, indicating that they have already class-switched and wouldn't be able to give rise to IgM. How do the authors reconcile these two findings?

8) Figure 3C, it would be very informative to show the frequency of clones that are clonally expanded within each ASC cluster, especially in ASC_Late_1 and ASC_Late_2.

9) In Figure 4B, please include appropriate statistical analysis to support the claims in lines 240-241.

10) In figure 5, please show the frequency of clones clonally expanded within each MemB sub-cluster.

11) In Figure 6, the authors elegantly used a S1pr2-creERT2-TdTomato-Prdm1-EYFP transgenic mouse to confirm that the B cell memory is indeed GC-derived. It would be very insightful to also confirm the GC origin or not of the clusters ASC_Late_1 and ASC_Late_2 using this model. As ASC_Late_2 is mainly IgM, it could be that this population is not arising from GC reactions.

12) In lines 275 to 281, the authors describe and define the different Memory B cell clusters. According to these lines, MemB_1 is defined as recent GC-derived and MemB_3 corresponds to MemB as activated cells. However, in Figure 6, the authors focused on MemB_2 defined as Fcrl5+ and MemB_4 as CD23+, which are defined as atypical Bmem and Bmem depending on T cell for their activation respectively. Why did the authors choose to focus only on these two clusters? Did the authors confirm that MemB_1 and MemB_3 have a GC origin?

13) In Figure 6E, can authors include GL7 staining to exclude that labeled S1pr2+ and Fcrl5+ cells don't belong to the GC compartment. Also, quantification of the confocal images is required.

---

## [Author Response]

Essential revisions:Both of the reviewers recognize the potential interest of this manuscript but have raised several concerns.1) The authors should perform additional experiments to validate whether the memory B cell subsets identified using Fcrl5 and CD23 are spatially distinct. These experiments should include quantification of the imaging data and a revised flow cytometry gating strategy to identify memory B cell subsets. The imaging data will likely require an improvement in the quality of the staining to better identify the Fcrl5+ cells.

We agree with the reviewers that a thorough confirmation of spatial localization through quantifiable microscopy would be ideal for clarifying the distribution of the different Memory B cell subsets. Unfortunately, the sparsity of these cells, in addition to the suboptimal performance of the anti-FCRL5 antibody in microscopy limited our capacity to do so. However, it was the observation of fate mapped cells in the Marginal zone area that led us to perform the short-term anti-CD45 i.v. injection labelling as a high throughput method for validation. Rapid staining with i.v. injected anti-CD45 antibody before organ extraction is a well-documented method to contrast localization between Marginal and Follicular zone B cells through flow cytometry, based on the fluorescent labelling of CD45(Cinamon et al., 2008). Indeed, Song et al. (2022) recently used this approach in support to the localization of a B cell population akin to our MemB_2 with a Tbet-reporter mouse after infection and found similar results with flow cytometry and imaging.

To support this approach, we are including flow cytometry data from the same samples using a traditional CD21 vs CD23 gating strategy to distinguish Marginal Zone and Follicular B cells, demonstrating the capacity to distinguish cells in these compartments. Importantly, using this traditional gating strategy we observe S1PR2+ fate mapped cells in the Marginal zone compartment; the concept of memory in the Marginal Zone remains largely unexplored and we hope our publication will support this idea.

We are including now all the FMOs for Figure 6 and the confirmation of i.v. staining as supplementary Figure 6- supplement 1 in the revised manuscript.

2) The authors should add additional discussion regarding how their memory B cell subsets related to subsets that have been previously identified in mice using markers such as CD80 and PDl2.

We are aware of previous studies using CD80 and PDL2 for characterizing the Memory B cell compartment. Interestingly, we didn’t observe much expression of those genes in the clusters that we decided to focus on, MemB_2 and MemB_4, but did find them expressed for clusters MemB_1 and MemB_3, especially in the immunized mice (Author response image 1). These markers were first defined in the context of immunization and by specific staining with fluorophore labelled antigen and IgG1 positivity. We hypothesize that the distinct conditions of our model and the fact that we took the full Memory B cell compartment in consideration might account for the difference in expression pattern.

**Author response image 1. sa2fig1:** Normalized gene expression of *Cd80* (left) and *Pdcd1lg2*(coding for PD-L2, right), split by MemB cluster and condition.

3) One of the immunization models used in this study relies on a secondary dose of NP-OVA to activate previously labeled GC-derived B cells. This acute system differs from the chronic B cell activation observed in the autoimmune disease model, in which labeled B cells can be constantly selected into continually ongoing and spontaneous GC and differentiate into antibody-secreting cells. The authors should explain in some lines the differences between these two models and why they decide to compare an acute NP-immunization model with a chronic autoimmune disease model?

Our focus going into these experiments and in the development of this manuscript was to explore the diversity of wild type B cells developing in the autoimmune environment and to understand their clonal intersects, much of which was still unknown or understudied in the field. It was of course important for us to “anchor” our observations into a more traditional scheme, so that our data could be of broader use. We decided to use a traditional immunization scheme with the widely used 4-Hydroxy-3-nitrophenylacetyl (NP) hapten coupled to Ovalbumin antigen as reference. We hope it can be appreciated that because we understand the major differences between these models, we refrain ourselves from major comparisons like differential expression analysis between conditions. Though we have explored those analysis we believed them to be outside the scope of this article. We propose that a more complete exploration of the system would likely require the usage of both acute and chronic foreign antigen presence as contrast. Indeed, recent analysis of chronic or severe viral infection are showing overlaps with self-reactive immune response characteristics (Woodruff et al., 2020, 2022).

4) The model of tamoxifen administration used in figures 1 to 5 (two doses of tamoxifen on days 4 and 7 after initial immunization) and then in figure 6 (tamoxifen diet for 12 weeks) are different. This might result in a different labeling pattern for the cells expressing AID or S1pr2-Cre. Have the authors made sure that both tamoxifen administration schedules are comparable in terms of cell labeling output?

We apologize for the confusion, as for the validation experiments demonstrated in Figure 6 in the manuscript, we used a similar strategy (diet) for tamoxifen induction in both *Aicda*- and *S1pr2*-Cre mice. We had a typo in the original manuscript, with the *Aicda* reporter validation done with tam diet for 8 weeks and left out of diet for 4 weeks, and all *S1pr2* reporter experiments done with a tam diet of 12 weeks. This is corrected in line 247 from the revised manuscript and displayed in the figures as well. Our goal with the tamoxifen diet was to maximize the number of cells that we could fate map since as soon as possible after the chimeras are produced, in contrast to acute labeling by injection of gavage. We have been able to observe the MemB populations in all scenarios, tamoxifen diet and injection with both Cre reporters.

5) In Figure 2, the authors identified two terminally differentiated clusters, ASC_Late_1 and ASC_Late_2, which display different transcriptomics. Using differentially expressed markers, can the authors identify these two ASC populations by flow cytometry as performed in Figure 6 for memory B cells?

We agree that further validation of these populations is granted, unfortunately the markers we pursued didn’t allow complete cluster discrimination through flow cytometry. We hypothesize this happened because most of the cluster gene markers for ASCs didn’t appear as positive or negative but represented different degrees of expression.

We show here *Itga4* (CD49d) as an example (Author response image 2), where we explored the staining for CD49d in our 564:*S1pr2* chimeras, within the S1PR2+ compartment, by conventional markers for B cells (B220+ CD138-), Plasmablasts (B220+ CD138+) and Plasma cells (B220- CD138+), confirming we can identify CD49d lo and hi populations in the Plasma cells as suggested by the single cell data, but these populations were also found in the B220+CD138+ gate, traditionally considered a less mature phenotype, complicating the marker usage interpretation.

**Author response image 2. sa2fig2:** Flow cytometry profile for CD49d expression for a representative 564:*S1pr2* chimera, showing the CD49d lo an CD49 hi populations within the B cell, Plasmablast and Plasma cell conventional gates.

6) In figure 3A, ASC_Late_1 has a very diverse isotype usage in the auto-immune mice but not in the immunized ones. On the contrary, ASC_Late_2 in both conditions is overwhelmingly dominated by IgM. Are authors convinced that both ASC populations arise from the autoimmune disease? Are the antibodies generated by both ASC subsets specific for auto-antigens?

We hypothesize that the discrepancy in isotype diversity between conditions (Immunized vs Autoimmune) more likely arises from the distinct antigenic exposure in the model. In the autoimmune mice there is likely recurrent exposure and ongoing response to a diversity of antigens, which leads to a larger engagement of cells into isotype switch, whereas the immunization strategy engages a more limited number of cells, most of which don’t seem to have switched yet, because of its acute and highly specific nature, and likely lesser degree of antigenicity and re-engagement of cells from the primary response.

7) In Figure 3A, isotype usage for each ASC cluster and in both conditions is shown. According to previous pseudo-time analysis, ASC_Late_2, which mainly displays an IgM BCR, is originated from ASC_Early_1 to ASC_Early_2 to ASC_mild_1 to ASC_mild_2 according to the pseudo-time analysis (figure 2B). However, all these progenitors' clusters display a much lower IgM expression, especially in the auto-immune model, indicating that they have already class-switched and wouldn't be able to give rise to IgM. How do the authors reconcile these two findings?

Our intention with the Pseudotime analysis was to use an unsupervised method to find likely terminal stages in ASC development and support our assignment, which was originally driven by already known markers. We would like to note that the interpretation of Pseudotime analysis can be tricky and should be taken with caution. For example, the ‘time’ transitions are unfortunately not as clear cut and a few Mid_3 cells will be represented in Pseudotime_2 leading to ASC_2, and even a few cells from Late_2 can show in Pseudotime_1. Nevertheless, a useful way to understand Pseudotime trajectories is to think about it as a transportation system where every cell cluster is a stop. It is possible then that IgM individuals follow along the system but get out mostly at the cluster ASC_Late_2 stop. Indeed, we observe an interesting distribution in isotype usage when splitting the cells into those that were considered only for Pseudotime_1 or Pseudotime_2, with an enrichment of IgM cells already in ASC_Mid_3 stop (Author response 3).

**Author response image 3. sa2fig3:** Isotype usage per ASC cluster, showed only for cells assigned to Pseudotime_1 (left) or Pseudotime_2 (right).

8) In lines 275 to 281, the authors describe and define the different Memory B cell clusters. According to these lines, MemB_1 is defined as recent GC-derived and MemB_3 corresponds to MemB as activated cells. However, in Figure 6, the authors focused on MemB_2 defined as Fcrl5+ and MemB_4 as CD23+, which are defined as atypical Bmem and Bmem depending on T cell for their activation respectively. Why did the authors choose to focus only on these two clusters? Did the authors confirm that MemB_1 and MemB_3 have a GC origin?

We decided to follow up on MemB_2 and MemB_4 because Fcrl5 and Cd23, two surface markers that characterize those clusters, are easily identifiable by flow cytometry and have traditional spatial distribution connotations in the context of B cell biology: Cd23 being a conventional marker for follicular B cells and Fcrl5 having been linked to the Marginal Zone compartment. It was interesting to us that such markers distinguished two populations of Memory B cells *after* leaving the GC, as distinct localization likely suggests distinct function. Moreover, FCRL5 allows us to us to link our interpretations to the work of other groups investigating the nature of “atypical” B cells in mice and human (DN2-like) and as a demonstration of that phenotype presence in our model. Importantly, as mentioned in the original manuscript, we believe this to be a confirmation of a DN2-like population within the GC-derived memory compartment. We expect this finding to be of high relevance to ongoing discussions on the nature of these cells within the autoimmune and B cell field at large as so far this phenotype has mostly been linked to recent activation from naïve B cells outside the GC.

We are of course interested in the other clusters as well but decided to constrain ourselves to two of them to maintain our research focused, as we learned that single cell rna-seq validation is on itself a daunting task to fit within a single publication.

Reviewer #1 (Recommendations for the authors):Specific points1) The scheme in Figure 1A does not include the exact time points in which mice received tamoxifen and are then sacrificed. Could the authors add this information to the figure for clarity?

Yes, we have now included that information.

2) Figure 1D is missing statistical analysis to support the claims of the authors in lines 87-88-113.

We originally decided not to include statistical support given the difficulties to do proper differential abundance in single cell rna-seq data, we have modified the text to reflect this. Though we have observed it being used in other articles, the approach to differential abundance has not yet matured as much as the differential expression in single cell rna-seq data.

3) Figure 1E displays the phylogenetic tree of the most expanded clone in one of the autoimmune chimeras. The authors claim that expanded clones can be found in all 4 clusters, but looking closely at this figure the MemB cluster is scarcely represented. Previous work (Viant et al. 2020) showed that the memory B cell compartment is largely populated by single cell clones rather than expanded clones. Can the authors quantify the clonal expansion of memory B cells in their model (how many single-cell, 2-cell, 3-cell clones?).

To clarify, our intention with that expression and figure was to show that clonal members can be present in all compartments, but this does not mean that these represent highly expanded clones in each compartment. We have now tried to make this evident in the text. Indeed, we do find many singletons when analyzing each compartment separately. We would like to point that negative results (e.g. absence or limited clonal members) have a restricted interpretation in single cell repertoire given the limited number of cells that one is capable to query so far, even when droplet-based methods allow to sequence more cells that plate sorted methods do; and because of the dominance of cells with high immunoglobulin transcript count (ASCs) in the data tend to dominate over MemB and GC cells when being part of the same sequencing library.

4) In figure 1I, the authors show that two monoclonal antibody pools originating from autoimmune chimeras could bind to some auto-antigens from a human recombinant protein array. Why did the authors check the binding to human autoantigens and not mouse autoantigens? Could they be missing some targets that do not cross-react between mouse and humans?

We agree with this observation and whole-heartedly wish we could have done otherwise. It is however not really a decision in the autoimmune B cell field to work with human proteins, as mouse reagent counterparts in that scale don’t exist. This is still a major limitation in the field.

5) In Figure 2, the authors identified two terminally differentiated clusters, ASC_Late_1 and ASC_Late_2, which display different transcriptomics. Using differentially expressed markers, can the authors identify these two ASC populations by flow cytometry as performed in Figure 6 for memory B cells?

Please see Essential revisions question 5.

6) In figure 3A, ASC_Late_1 has a very diverse isotype usage in the auto-immune mice but not in the immunized ones. On the contrary, ASC_Late_2 in both conditions is overwhelmingly dominated by IgM. Are authors convinced that both ASC populations arise from the autoimmune disease? Are the antibodies generated by both ASC subsets specific for auto-antigens?

Please see Essential revisions question 6.

7) In Figure 3A, isotype usage for each ASC cluster and in both conditions is shown. According to previous pseudo time analysis, ASC_Late_2, which mainly displays an IgM BCR, is originated from ASC_Early_1 to ASC_Early_2 to ASC_mild_1 to ASC_mild_2 according to the pseudo-time analysis (figure 2B). However, all these progenitors' clusters display a much lower IgM expression, especially in the auto-immune model, indicating that they have already class-switched and wouldn't be able to give rise to IgM. How do the authors reconcile these two findings?

Please see Essential revisions question 7.

8) Figure 3C, it would be very informative to show the frequency of clones that are clonally expanded within each ASC cluster, especially in ASC_Late_1 and ASC_Late_2.

We agree on the importance of this information but were not able to find a better way to visualize the intersects while adding properly sized numerical values. The data for these circos plots is of course provided in the Supplementary Tables.

9) In Figure 4B, please include appropriate statistical analysis to support the claims in lines 240-241.

Please see response to question 2 on this section.

10) In figure 5, please show the frequency of clones clonally expanded within each MemB sub-cluster.

Please see response to question 8 on this section.

11) In Figure 6, the authors elegantly used a S1pr2-creERT2-TdTomato-Prdm1-EYFP transgenic mouse to confirm that the B cell memory is indeed GC-derived. It would be very insightful to also confirm the GC origin or not of the clusters ASC_Late_1 and ASC_Late_2 using this model. As ASC_Late_2 is mainly IgM, it could be that this population is not arising from GC reactions.

We appreciate the reviewer interest on this aspect, as it represents a fundamental question to the ASC field we are actively pursuing. Indeed, one of our long-term goals is the distinction of GC and Extrafollicular contributions to the ASC compartment that we think the reviewer is alluding to. Unfortunately, we cannot yet answer that question with this data and system. For example, let’s say that a cell emerges from the GC as a fate mapped MemB, but the transition to an ASC happens directly (MemB-to-ASC) in the Extrafollicular space, without returning to the GC; that ASC is by localization non-GC derived but would be GC-derived by lineage, this distinction is clouded in the long-term fate mapping approach and requires more detailed kinetics and spatial studies. We are working on a modification on the experimental design to address this precisely.

12) In lines 275 to 281, the authors describe and define the different Memory B cell clusters. According to these lines, MemB_1 is defined as recent GC-derived and MemB_3 corresponds to MemB as activated cells. However, in Figure 6, the authors focused on MemB_2 defined as Fcrl5+ and MemB_4 as CD23+, which are defined as atypical Bmem and Bmem depending on T cell for their activation respectively. Why did the authors choose to focus only on these two clusters? Did the authors confirm that MemB_1 and MemB_3 have a GC origin?

Please see Essential revisions question 8.

13) In Figure 6E, can authors include GL7 staining to exclude that labeled S1pr2+ and Fcrl5+ cells don't belong to the GC compartment. Also, quantification of the confocal images is required.

We used CD169, delineating the Marginal Zone through macrophage staining, as a reference to localization. As shown in the images provided, this allows us to clearly distinguish the boundaries of the follicles and can be confident these don’t represent GC B cells (which can be seen in red clusters within the follicle).